# Maternal Folate Status and the Relation between Gestational Arsenic Exposure and Child Health Outcomes

**DOI:** 10.3390/ijerph191811332

**Published:** 2022-09-09

**Authors:** Marisa A. Patti, Karl T. Kelsey, Amanda J. MacFarlane, George D. Papandonatos, Tye E. Arbuckle, Jillian Ashley-Martin, Mandy Fisher, William D. Fraser, Bruce P. Lanphear, Gina Muckle, Joseph M. Braun

**Affiliations:** 1Department of Epidemiology, Brown University, 121 S Main St., Providence, RI 02903, USA; 2Nutrition Research Division, Health Canada, 251 Sir Frederick Banting Driveway, Ottawa, ON K1A 0K9, Canada; 3Department of Biology, Carleton University, 1125 Colonel By Dr., Ottawa, ON K1S 5B6, Canada; 4Department of Biostatistics, Brown University, 121 S Main St., Providence, RI 02903, USA; 5Environmental Health Science and Research Bureau, Healthy Environments and Consumer Branch, Health Canada, 50 Colombine Driveway, Ottawa, ON K1A 0K9, Canada; 6Department D’obstétrique et Gynécologie, Université de Sherbrooke, 2500 Bd de L’Université, Sherbrooke, QC J1K 2R1, Canada; 7Department of Health Sciences, Simon Fraser University, 515 W Haastings St., Vancouver, BC V5A 1S6, Canada; 8School of Psychology, Université Laval, Ville de Québec, 2325 Rue de L’Université, Québec, QC G1V 0B4, Canada

**Keywords:** neurodevelopment, anthropometry, phenome, prenatal, arsenic, folate, folic acid

## Abstract

Gestational arsenic exposure adversely impacts child health. Folate-mediated 1-carbon metabolism facilitates urinary excretion of arsenic and may prevent arsenic-related adverse health outcomes. We investigated the potential for maternal folate status to modify associations between gestational arsenic exposure and child health. We used data from 364 mother–child pairs in the MIREC study, a prospective pan-Canadian cohort. During pregnancy, we measured first trimester urinary arsenic concentrations, plasma folate biomarkers, and folic acid supplementation intake. At age 3 years, we evaluated twelve neurodevelopmental and anthropometric features. Using latent profile analysis and multinomial regression, we developed phenotypic profiles of child health, estimated covariate-adjusted associations between arsenic and these phenotypic profiles, and evaluated whether folate status modified these associations. We identified three phenotypic profiles of neurodevelopment and three of anthropometry, ranging from less to more optimal child health. Gestational arsenic was associated with decreased odds of optimal neurodevelopment. Maternal folate status did not modify associations of arsenic with neurodevelopmental phenotypic profiles, but gestational arsenic was associated with increased odds of excess adiposity among those who exceed recommendations for folic acid (>1000 μg/day). However, arsenic exposure was low and folate status was high. Gestational arsenic exposure may adversely impact child neurodevelopment and anthropometry, and maternal folate status may not modify these associations; however, future work should examine these associations in more arsenic-exposed or lower folate-status populations.

## 1. Introduction

In utero exposure to arsenic, a ubiquitous and toxic metalloid [1], may induce adverse effects on child neurodevelopment [2,3,4,5,6] and anthropometry [7,8,9,10,11]. Folate facilitates urinary excretion of arsenic through folate-mediated 1-carbon metabolism, glutathione conjugation, or reductive methylation [12,13,14,15,16,17]. For this reason, folic acid supplementation has been used as a dietary intervention to reduce bodily burdens of arsenic to prevent disease in adults [2,12]. However, less is known about the potential of gestational folate status to modify associations between fetal arsenic exposure and child health. 

Some studies have found evidence to suggest that folate may modify associations between environmental exposures and child health, although not all reports concur. Pesticides, air pollution, and phthalates are environmental exposures previously identified for their association with atypical neurodevelopment; folic acid intake has been shown to modify their associations with Autism Spectrum Disorder (ASD) [18,19,20,21], and cognitive abilities [22]. However, prior work found limited evidence to suggest dietary folate intake affected the relationship between low-level urinary arsenic concentrations and child academic achievement or cognition [23,24]. 

Most of the relevant studies have evaluated the impacts of gestational arsenic exposure and folate status on child neurodevelopment separately from their effects on anthropometry. However, there is an interest in understanding how environmental exposures influence the patterns and profiles of multiple aspects of child health, i.e., the ‘phenome’ [25,26,27,28]. A phenome approach has been increasingly adopted in psychology, as research moves away from distinct psychiatric disorders classified within the Diagnostic and Statistical Manual of Mental Disorders (DSM) and towards Research Domain Criteria (RDoC), because biobehavioral features often overlap with heterogeneously defined disorder categories [26]. For example, previous editions of the DSM may not have captured the true spectrum of behaviors present amongst individuals with ASD or attention deficit hyperactivity disorder (ADHD) [29,30]. The presence of co-occurring conditions can create additional barriers, beyond those presented by individual health concerns. Thus, evaluating multi-morbidity may lead to greater insight about the totality of health effects related to arsenic exposure.

Using data from a prospective cohort study of Canadian maternity patients, we evaluated whether maternal plasma folate concentrations and folic acid intake modified the relation of gestational urinary arsenic concentrations, using individual measures and phenotypic profiles of child neurodevelopmental and anthropometric health outcomes. We hypothesized that the adverse effects of gestational arsenic on phenotypic profiles of child health outcomes at age 3 years will be diminished in children whose mothers had higher folate levels.

## 2. Materials and Methods

### 2.1. Study Participants

We used data from the Maternal–Infant Research on Environmental Chemicals (MIREC) study. MIREC is a pan-Canadian prospective cohort study designed to assess the impact of environmental chemical and nutritional exposures on maternal, infant, and child health, with recruitment procedures previously described [31].

Briefly, pregnant women in their first trimester were recruited from 10 cities (11 sites) across Canada from 2008–2011. Eligibility criteria including ≥18 years of age, <14 weeks of gestation, willing to provide a sample of cord blood, planning on delivering at a local hospital, and no history of known fetal abnormalities or serious medical complications. Of 8716 women invited to participate in MIREC, 2001 consented to participate, and 1861 delivered singleton live births. Women provided biospecimens and completed questionnaires related to sociodemographics, lifestyle, and nutritional factors during pregnancy. In-person follow-up at around age 3 years was completed at six cities on 610 singleton children. Child neurodevelopmental and anthropometric measures were evaluated during follow-up visits at age 3 years, and complete outcome information was available for 501 participants. Of these, 409 participants had available data detailing first trimester plasma folate biomarkers, folic acid intake from supplements, and maternal urinary arsenic concentrations, and had consented to participation in the MIREC Biobank. After excluding those with incomplete covariate data (*n* = 45), our final analytic sample included 364 mother–child dyads (Appendix A). 

The MIREC Study and the MIREC Biobank were approved by the Health Canada, and Public Health Agency of Canada Research Ethics Board, and all participating study sites’ ethics review committees. All women provided written informed consent for themselves and their participating children. 

### 2.2. Maternal Urinary Arsenic Concentrations 

Trained staff collected spot urine samples during the first trimester of pregnancy; these were frozen at ≤−20 °C within 2 h of collection and shipped on dry ice to the MIREC coordinating center in Montreal, where they were stored at ≤−30 °C thereafter [32]. No contamination was detected during the collection, processing, transportation, or storage of these samples [32]. Arsenic species concentrations (arsenite [As-III], arsenate [As-V], dimethylarsinic acid [DMA], monometthylarsonic acid [MMA], and arsenobetaine [AsB]) in these samples were quantified at the Centre de Toxicologie du Québec, Institut national de santé publique du Québec [32]. MIREC project laboratories participate in regular quality assurance committee meetings to ensure compliance with internal standards.

Concentrations were measured by inductively coupled plasma mass spectrometry (HPLC coupled with Varian 820 MS ICP-MS) after urine samples were diluted (1/10) with solvent. The limit of detection (LOD) for this method was 0.75 µg As/L (0.01 µmol/L) [33]. To assess urine dilution, trained research staff measured specific gravity in thawed urine samples using a handheld refractometer. Arsenic biomarker values below the LOD were replaced by LOD/√2. Urinary arsenic concentrations were standardized for specific gravity to account for urine dilution using a previous formula [33,34], and were log_10_-transformed to reduce the influence of outliers. Note that only standardized specific gravity values of urinary arsenic species concentrations were considered for statistical analysis. 

Over 80%, 85%, and 93% of samples, respectively, had As-V, As-III, and MMA concentrations below the LOD, and thus were excluded from subsequent analyses. Given that roughly half (51%) of samples had AsB concentrations below the LOD, and AsB is considered a non-toxic arsenic species most commonly found in seafood, we excluded it from subsequent analyses [35,36]. Thus, we considered only first trimester urinary DMA concentrations as the primary arsenic exposure variable for primary and secondary analyses.

### 2.3. Maternal Folate Status

#### 2.3.1. Maternal Plasma Folate Status 

Blood samples were collected by venipuncture up to twice during pregnancy (first and/or third trimester) at scheduled clinic visits. Only first trimester values are considered here, in order to maintain temporal alignment with arsenic exposure. Blood was collected into vacutainers coated with ethylenediaminetetraacetic acid (EDTA). The blood was centrifuged within 15 min of collection, and plasma was immediately aliquoted and stored at −80 °C. Samples from the MIREC Biobank were shipped to Health Canada and stored at −80 °C until analysis. 

We determined the contributions of individual folate vitamers to plasma total folate using liquid chromatography–tandem mass spectrometry (LC-MS/MS) at the Health Canada Food Laboratory (Toronto, ON, Canada), using methods defined by the US Centers for Disease Control and Prevention and modified for manual injection [37,38,39]. Folate vitamers included tetrahydrofolate (THF), 5,10-methenylTHF, 5-formylTHF, 5-methylTHF, UMFA, and MeFox, 4α-hydroxy-5-methylTHF, which is the oxidation product of 5-methylTHF. Total folate was calculated by summing the concentrations of 5-methylTHF, folic acid, THF, 5-formylTHF, and 5,10-methenylTHF. 

Plasma samples were spiked with an internal standard mixture followed by addition of the extraction buffer (at pH 5), and incubated at 4 °C for 30 min. Sample extraction and cleanup was performed by solid-phase extraction, where final extracts were filtered prior to LC-MS/MS analysis. Folate peaks were separated by UPLC gradient mobile phase and detected by MS/MS in MRM mode. Quantitation was performed with external calibration standards based on peak area ratio using internal standard. Samples were analyzed using the Waters Xevo TQ-XS Mass Spectrometer coupled to the Waters Acquity UPLC system (Waters Limited, Mississauga, ON, Canada). Data collection and reduction was achieved using MassLynx software (V4.2 Waters Corporation, Milford, MA, USA). Total folate values were log_10_-transformed to satisfy normality assumptions.

#### 2.3.2. Maternal Folic Acid Intake from Supplements 

After the first trimester study visit, participants self-reported supplement and medication intake within the last 30 days [40]. Participants provided the product name and description, identification number, as well as the amount and frequency of intake. Using previously described methods, we estimated the total daily folic acid intake from all supplements [20]. 

We categorized total daily folic acid intake from supplements into three groups, i.e., those not meeting recommended folic acid intake (0 to <400 µg/day), meeting recommended folic acid intake (≥400 to ≤1000 µg/day), and exceeding recommended folic acid intake (>1000 µg/day) [37]. 

### 2.4. Child Neurodevelopmental and Anthropometric Health Outcomes 

Child neurodevelopmental assessments were carried out and anthropometric measurements taken at approximately 3 years old (range: 3.0–3.9 years; SD: 0.32) at the participants’ home or study clinic. Research staff were trained and monitored by a PhD clinical psychologist to assess child cognitive abilities using the Wechsler Preschool and Primary Scale of Intelligence (WPPSI-III) [41]. Behavior problems and social cognition were assessed by maternal report two weeks before or at the study visit, using the Behavioral Assessment System for Children (BASC-2) [42] and the Social Responsiveness Scale (SRS-2) [43]. 

Trained research assistants assessed anthropometry at or within six months of the neurobehavioral visit. Measurements included weight, height, head/waist/hip circumferences, and subscapular and triceps skinfold thickness. Length was measured with a portable stadiometer to the nearest 0.1 cm and weight using a digital scale to the nearest 2 g. We derived age- and sex-standardized child body mass index (BMI) z-scores using World Health Organization (WHO) standards [44]. Head, waist, and hip circumferences were measured using a measuring tape and established protocols [45]. Using standardized procedures (i.e., identifying landmarks where measurements can be taken), a caliper was employed to estimate skinfold thickness of the back (subscapular) and arm (triceps). Two measures were taken for all anthropometric measurements, except in cases where discrepancies were greater than a predetermined value (i.e., weight: 0.1 kg, height: 0.5cm, subscapular and triceps skinfold thickness: 2 mm), which required a third measure. 

#### Classes of Child Neurodevelopmental and Anthropometric Health Outcomes

We created neurodevelopmental and anthropometric health phenotypic profiles using six neurodevelopmental outcomes: WPPSI full scale intelligence quotient (FSIQ), SRS T-scores, four BASC composite scores (externalizing problems, internalizing problems, behavioral symptoms index [BSI], and adaptive skills), and six anthropometric outcomes (BMI, head circumference, hip circumference, waist circumference, triceps skinfold thickness, and subscapular skinfold thickness). To account for varying scales and directions of neurodevelopmental and anthropometric outcomes, we reverse-scored selected outcomes and converted all scores and measures to z-scores.

We used latent profile analysis (LPA) to identify separately the phenotypic profiles of childrens’ neurodevelopmental and anthropometric health outcomes, referring to the latent classes of neurodevelopmental and anthropometric health outcomes as phenotypic profiles. We assumed equal variance and zero covariance between individual health measures when conducting LPA. Based on the LPA results, we identified three phenotypic profiles of neurodevelopment and three phenotypic profiles of anthropometry, respectively (Appendix A). 

Children were characterized with non-optimal, typical, and optimal scores based on neurodevelopmental outcomes. Non-optimal phenotypic profiles of neurodevelopment (*n* = 31) were characterized by lower FSIQ and BASC adaptive skills scores than average, as well as higher average SRS, BASC externalizing, BASC internalizing, and BASC BSI scores, suggesting lower average cognitive abilities, less adaptive behavior, reduced reciprocal social behavior, and more problem behavior (Appendix A). Typical (*n* = 183) and optimal (*n* = 150) phenotypic profiles of neurodevelopment were characterized by average or higher than average FSIQ scores, respectively, and lower than average SRS, BASC adaptive skills, externalizing, internalizing, and BSI scores. We identified three phenotypic profiles of anthropometry that we characterized as low, average, and excess adiposity. Children in low (*n* = 41), average (*n* = 218), and excess (*n* = 105) adiposity phenotypic profiles were characterized by lower, average, and higher than average BMI, waist and hip circumference, and also triceps and subscapular skinfold thickness measures, respectively (Appendix A). Average head circumference values were similar between the low and average adiposity phenotypic profiles, and were only slightly higher for the excess adiposity phenotypic profile.

### 2.5. Covariates

Covariate information was collected via maternal self-report using questionnaires completed during the first trimester (baseline), as well as during the home or clinic follow-up visit, when children were 3 years old. Mothers reported perinatal information, including parity and pre-pregnancy BMI, on standardized questionnaires during the baseline visit. Plasma cotinine, measured during the first trimester, was used as a biomarker of smoking and second-hand tobacco smoke exposure [46,47]. Study staff assessed the quality and quantity of the caregiving environment using the Home Observation for Measurement of the Environment (HOME) inventory at the 3 years visit [48]. The Center for Epidemiologic Studies depression scale (CES-D10) was used to assess maternal depressive symptoms [49]. 

### 2.6. Statistical Analyses

First, we described study sample characteristics and distributions of DMA and plasma folate values as covariates. Next, we calculated Spearman correlations for DMA and plasma folate vitamers, and for DMA and categories of folic acid intake from supplements. 

We selected covariates for adjustment using a directed acyclic graph (DAG; Appendix A). We adjusted for maternal age (continuous), maternal country of origin (Canadian born or foreign born), prenatal vitamin use (yes or no), annual household income (continuous), pre-pregnancy BMI (normal and underweight, overweight, obese), maternal depressive symptoms (continuous CES-D scores), and parity (nulliparous, one previous child, two or more previous children). 

We used multivariable multinomial regression to estimate the covariate-adjusted differences in odds of neurodevelopmental or anthropometric phenotypic profile assignment, according to first trimester urinary DMA concentrations. The typical neurodevelopmental phenotypic profile and average adiposity anthropometric phenotypic profile served as the respective reference groups. 

Next, we explored whether total plasma folate values or folic acid intake from supplements modified the associations of first trimester urinary DMA concentrations with child neurodevelopmental and anthropometric health phenotypic profiles. We used separate multinomial regression models to estimate covariate-adjusted odds of phenotypic profile assignment according to first trimester urinary DMA concentrations, stratified by terciles of total plasma folate values as well as categories of folic acid intake from supplements. Given the small number of participants who did not meet daily recommendations for folic acid intake (*n* = 19, 5%), we restricted our analysis to those who met daily recommendations (reference group) or those who exceeded daily recommendations. We formally tested whether associations between first trimester urinary DMA concentrations and phenotypic profiles of child health were modified by maternal folate status, by using product interaction terms between folate status and urinary DMA concentrations.

#### Secondary and Sensitivity Analyses

We conducted several secondary and sensitivity analyses. First, we used linear regression to estimate the covariate-adjusted associations between first trimester urinary DMA concentrations and individual child health outcomes. Second, we conducted modification analyses to determine whether maternal folate status modified associations between DMA and individual neurodevelopmental and anthropometric health outcomes. 

Next, we evaluated the robustness of our findings with additional adjustment for quality of caregiving environment (HOME scores), maternal smoking during pregnancy (evaluated using first trimester plasma cotinine concentrations), and categories of folic acid intake from supplements (only in those models considering total plasma folate). 

To account for uncertainty in latent profile assignment [50,51], we conducted sensitivity analysis of survey-weighted multinomial regression analysis using the R package svyVGAM [52]. Given an optimal choice of K latent profiles, we replicated each individual participant’s covariate data K times and successively assigned each row the profile k = 1,…,K, with weight equal to the posterior probability of latent profile membership. In addition, a working independence correlation matrix was employed, with individual participant as the cluster identification.

We used R version 4.1.0 (R Core Team, Vienna, Austria) for all statistical analyses [53].

## 3. Results

Women in the MIREC study were generally >30 years of age at the time of delivery (78%), university educated (70%), and had high annual household incomes ≥$100,000 CAD/year (44%) (Table 1). Most participants were born in Canada (84%), did not smoke during pregnancy (70%), were nulliparous prior to enrollment (48%), and had normal or underweight pre-pregnancy BMIs (62%). During their pregnancies, 90% of participants reported taking prenatal vitamins. 

Median concentrations were lowest for As-V, followed by MMA, As-III, DMA, and AsB (Appendix A). First trimester urinary DMA concentrations were similar across study sample characteristics, and were only modestly elevated among those who did not meet daily recommendations for folic acid (Table 1). 

Median first trimester plasma folate vitamer concentrations were lowest for nonmethyl folates, followed by folic acid, and 5-methylTHF (Appendix A). Total plasma folate was higher with increasing age, maternal educational attainment, annual household income, folic acid supplementation, and lower parity (Table 1). Total plasma folate values were higher among mothers not born in Canada, and among those who took prenatal vitamins, while values were lower among those who smoked during pregnancy. 

The majority (~95%) of participants reported folic acid intake from supplements that met or exceeded the daily recommended dose of 400 μg/day, with a quarter of participants exceeding the daily recommended dose of 1000 μg/day for women of childbearing age with a low risk of neural tube defects [54,55] (Appendix A). Only ~5% of participants reported folic acid intake below the daily recommendation.

First trimester total folate values were uncorrelated with urinary DMA concentrations (Pearson correlation coefficients range: −0.05 to −0.02, Appendix A). Spearman rank correlation coefficients between folic acid intake and plasma total folate biomarkers were weakly positive (0.17), but uncorrelated with DMA (−0.03) (Appendix A). 

Across phenotypic profiles of neurodevelopment and anthropometry, we did not observe substantial variation in the distributions of first trimester urinary DMA concentrations, total plasma folate concentrations, or folic acid intake from supplements (Figure 1, Appendix A). 

Except for the optimal neurodevelopmental profile, urinary DMA concentrations were not associated with odds of neurodevelopmental or anthropometric phenotypic profile assignment. For example, every 10-fold increase in first trimester urinary DMA concentrations was associated with a decreased likelihood of the optimal (OR: 0.44; 95% CI: 0.19, 1.02) neurodevelopmental phenotypic profile compared to the typical neurodevelopmental phenotypic profile (Figure 2, Appendix A). Increasing first trimester urinary DMA concentrations were associated with decreased odds of assignment to the low adiposity phenotypic profile (OR: 0.42; 95% CI: 0.11, 1.63) relative to the average adiposity phenotypic profile, but the association was imprecise (Figure 2, Appendix A).

The association between urinary DMA concentrations and odds of being in a non-optimal neurodevelopmental phenotypic profile did not differ by terciles of total plasma folate concentrations (interaction term *p*-value = 0.80) or folic acid intake (interaction term *p*-value = 0.71) (Figure 3, Appendix A). Among those who met daily recommendations for folic acid intake, every 10-fold increase in first trimester urinary DMA concentrations was associated with decreased likelihood of assignment to the optimal (OR: 0.35; 95% CI: 0.13, 0.98) versus the typical neurodevelopmental phenotypic profile (interaction term *p*-value = 0.21).

We observed a decreased likelihood of assignment to the low adiposity phenotypic profile with higher total plasma folate, but an increase of assignment to the excess adiposity phenotypic profile compared to the average adiposity phenotypic profiles, although interaction terms were not significant (interaction term *p*-value = 0.17 and 0.62, respectively) (Figure 4, Appendix A). For example, within the highest tercile of total plasma folate, every 10-fold increase in gestational DMA was associated with decreased likelihood of assignment to the low adiposity phenotypic profile (OR: 0.11; 95% CI: 0.00, 8.37), but increased assignment to the excess adiposity versus the average adiposity phenotypic profile (OR: 1.86; 95% CI: 0.39, 8.97). Among those who exceeded daily recommendations for folic acid intake, every 10-fold increase in first trimester urinary DMA concentrations was associated with almost 15-times the likelihood of assignment to the excess adiposity phenotypic profile (95% CI: 1.38, 157) compared to the average adiposity phenotypic profile, although this association was very imprecise. 

Results from adjusted linear regression models were null for the association between DMA and individual child health outcomes, with the exception of BASC internalizing scores (note, higher scores indicate more internalizing problem behaviors; Appendix A). In this case, each 10-fold increase in first trimester urinary DMA concentrations (*β*: 3.2; 95% CI: 0.3, 6.1) was associated with higher BASC internalizing problems scores. 

We also considered the potential for folate status to modify the association between first trimester urinary DMA concentrations and individual child health outcomes. Overall, across terciles of total plasma folate and categories of folic acid intake we observed null associations between urinary DMA concentrations and individual child health outcomes, with the exception of BASC internalizing behaviors (Appendix A). The association between first trimester DMA and internalizing problem behaviors increased in strength across the first (β: 2.6; 95% CI: −2.5, 7.7), second (β: 3.6; 95% CI: 2.0, 9.2), and third tercile of total plasma folate (β: 4.0; 95% CI: −0.9, 9.0) (interaction term *p*-value < 0.01). 

Associations between first trimester urinary DMA and neurodevelopmental and anthropometric phenotypic profiles did not appreciably change with additional adjustment for folic acid intake from supplements, maternal plasma cotinine, or HOME scores (Appendix A). However, for neurodevelopmental phenotypic profiles, additional adjustment for folic acid intake from supplements and maternal plasma cotinine concentrations accentuated associations of DMA with non-optimal phenotypic profile assignment, while adjustment for caregiving environment scores reduced that likelihood (Appendix A). We observed increased likelihood of optimal compared to typical neurodevelopment, specifically among those who exceeded daily recommendations of folic acid intake with additional adjustment for HOME scores. 

Accounting for uncertainty in latent profile assignments did not substantially alter the pattern of results. Point estimates remained largely the same, with some increases and decreases in precision (Appendix A). 

## 4. Discussion

Using data from a pan-Canadian prospective cohort study of pregnancy, we evaluated whether maternal folate status modified the relation between first trimester urinary DMA concentrations and child neurodevelopment and anthropometry. We found little evidence that maternal folate status or folic acid intake modifies the associations between gestational urinary DMA concentrations and phenotypic profiles of child neurodevelopment. However, gestational DMA was associated with a lower likelihood of optimal neurodevelopment among all participants, and a higher likelihood of excess adiposity among those who exceeded daily folic acid intake recommendations, although associations were imprecise. 

Arsenic is a naturally occurring element to which humans are exposed primarily through contaminated food and drinking water [1,56]. Some studies have found evidence that low levels of gestational and early-life arsenic exposure adversely impact child neurodevelopmental and growth outcomes [7,57,58], including reduced cognition, [2,3,59] atypical behaviors, [5,6,60] preterm birth, and low birth weight [7,9,10,11]. In contrast, other cohort studies did not find evidence that low-level gestational arsenic exposure impacts child cognition, psychomotor development, language, problem solving, or behavior [61,62,63,64,65]. These discrepant findings can be attributed to differences in the timing of arsenic exposure (i.e., gestation or early life), the biomarkers used (i.e., blood-based versus urine), or how arsenic was parameterized (i.e., total arsenic or speciated arsenic) [66].

Folic acid supplementation prevents neural tube defects, and promotes both maternal and child health [67,68,69]. Nevertheless, roughly 75% of women of reproductive age in the USA and 25% in Canada do not meet daily recommendations of folic acid necessary to promote child health (≥400 μg/day) [68,70]. They represent a potentially vulnerable subgroup, such that folate-deficient individuals may also have a diminished capacity to eliminate arsenic and its toxic intermediates. However, very few participants from the MIREC study did not meet minimum daily recommendations for folic acid intake, and many in fact exceeded daily recommendations. While intake of folate is known to be essential, few studies have examined thresholds for protective effects. We found that with increasing maternal urinary concentrations of DMA, among children whose mothers exceeded daily folic acid intake levels, the likelihood of a child being assigned to the excess adiposity profile was almost 15 times that of assignment to the average adiposity profile. Other studies have associated excessive folic acid intake with obesity, insulin resistance, and adverse birth outcomes in animal and human studies [71,72,73]. The Institute of Medicine defined a tolerable upper intake level for folic acid of 1000 μg/day, based on observed associations between high folic acid intake and exacerbation or promotion of vitamin B12 deficiency-related neurodegeneration [74]. 

Given that folate metabolism facilitates the urinary excretion of arsenic [12,13,14,15], folic acid supplementation has been promoted to reduce the bodily burden of arsenic in adults [75,76]. However, the current minimum daily recommend intake (400 μg) of folic acid may not be sufficient to reduce burdens of arsenic or prevent arsenic-associated illness [75,77]. While arsenic concentrations in MIREC participants were similar to the general Canadian and US population, they were lower than in populations from other studies [78,79]. 

We hypothesized that higher maternal folic acid intake would be associated with optimal child health, given how folate status has previously been found to favorably mitigate the relation between environmental exposures (i.e., pesticides, air pollution, phthalates, and lead) and child health [18,19,20,21,22]. However, earlier research found little evidence to suggest that dietary folate modified the relation between low urinary arsenic concentrations (median concentration: 9.5–11.7 μg As/L) and child cognitive abilities or academic achievement [23,24]. Among MIREC participants, first trimester urinary DMA concentrations were not associated with anthropometric phenotypic profiles, but were modestly associated with decreased likelihood of optimal neurodevelopment. When stratified by folate status, we did not observe evidence that total plasma folate or folic acid intake modified the relation between DMA and neurodevelopmental phenotypic profiles. Contrary to our hypothesis, among mothers who exceeded daily recommendations for folic acid, increasing DMA was associated with an increased likelihood of the child’s assignment to the excess adiposity phenotypic profile versus the average adiposity phenotypic profile. This observation will require further investigation in larger samples, because confidence intervals were large and imprecise. We speculate that these findings might be due to the low levels of urinary arsenic species and high levels of folate among MIREC participants. 

Our study has its strengths. The MIREC study provided rich data that allowed us to consider multiple child health outcomes, as well as maternal, reproductive, and child-level covariates. Additionally, the prospective study design of the MIREC cohort allowed for the assessment of gestational urinary arsenic concentrations and maternal folate status in association with subsequent child-health outcomes. Arsenic has previously been associated with adverse health outcomes in adults, including cancer, skin lesions, and cardiovascular disease [80]. This study provides a unique opportunity to study health outcomes in children following gestational exposure. We applied a phenome approach to evaluate child health holistically, recognizing the multi-morbidity that occurs in neurodevelopment and anthropometry [25]. Furthermore, we considered speciated arsenic as our primary exposure of interest. Given the differences in toxicity among arsenic intermediates, evaluating total arsenic alone would not allow for the nuanced analysis necessary to distinguish between the effects of specific species of arsenic intermediates on child health [36,81]. Additionally, we used two measures of maternal folate status, i.e., total plasma folate concentrations and folic acid intake from supplements, which were both collected during the first trimester of pregnancy. 

Our study also had some limitations. We observed relatively low urinary arsenic concentrations; thus, it is possible that we did not have a wide enough range of arsenic exposure to detect associations with health outcomes. While the health effects of low-dose arsenic exposure on child health may be inconclusive, it is important to recognize that there is no “safe” threshold of arsenic intake [82]. Additionally, given the large number of participants without detectable concentrations of MMA, As-V, and As-III, we were limited to considering only DMA, the least toxic arsenic intermediate [81]. It is possible that other health outcome associations may exist for other more toxic arsenic metabolites, along with a potential modifying effect of folate on any association between arsenic and child health. Further, folate and folic acid are involved in the methylation of arsenic from more toxic intermediates (i.e., MMA) to DMA so that it can be excreted from the body, thus DMA may prove more accurate than bodily burden for estimating arsenic excretion [81]. Assessing urinary concentrations of DMA without MMA could be inadequate for understanding the full extent to which folate may be involved in arsenic-related detoxification pathways [23,83]. Urinary arsenic concentrations can also be subject to misclassification, as only one measure of arsenic status may not accurately assess non-persistent exposure, due to its episodic nature [84]. Furthermore, urinary concentrations of DMA have low reproducibility throughout pregnancy, which could in part be due to changes in methylation efficiency attributed to physiological changes (i.e., kidney function) in pregnancy [85,86].

While we considered multiple neurodevelopmental assessments and anthropometric measures, our assessment of the child health phenome is incomplete. Future work may consider other known physical health conditions previously implicated with atypical neurodevelopment, such as seizure disorders [87], sleep problems [88,89], gastrointestinal disturbances [90,91], or childhood immune function [92]. Furthermore, all class assignments derived using LPA were determined based on the individual outcomes within this cohort and may not be representative of other populations. For example, the assigned phenotypic profile labels (i.e., non-optimal, typical, and optimal) were not based on clinically meaningful cut-points. In addition, the MIREC study was based in only one country, Canada, which may limit the external validity of our study for other regions. It is also possible that misclassification occurred among the outcome definitions. For example, young children do not fully exhibit internalizing behaviors, contributing to less reliable assessments at younger ages [93]. Also, child development is not a linear process, and a child with noted delays in early childhood may not sustain these into later childhood or adulthood. Thus, future work would benefit from evaluating repeated measures of child neurodevelopment and anthropometry, taken from early childhood through adolescence. 

While we leveraged a large prospective data source, our sample size was insufficient to detect small associations or modifications. Children in the MIREC study are generally of higher socioeconomic position and are healthy, as evidenced by the small proportion of children in the sample with non-optimal measures. This may have limited our ability to identify those children more vulnerable to the effects of multi-morbidity. Additionally, participating mothers were also generally healthy, and relatively few experienced pregnancy-related health concerns such as gestational diabetes, hypertension, or pre-eclampsia. Thus, we were unable to account for the impact of maternal health outcomes on our assessment of multimorbidity in children, though this should be considered in future work. 

Also, it is possible that misclassification occurred among plasma folate biomarkers or folic acid intake levels. Folic acid intake from supplementation was only collected by maternal self-report during the first trimester, which may not have reflected intake during the rest of pregnancy, and/or have been impacted by reporting bias. Additionally, maternal plasma folate samples were non-fasting, which could contribute to higher within-participant variability. It is important to note that in Canada, over-the-counter prenatal vitamins and supplements can contain up to the upper tolerable intake level of 1000 μg folic acid/day, the most common dose in prenatal supplements, though higher dose supplements are available by prescription [94]. Most MIREC participants consumed 1000 μg folic acid/day, which may have limited the analysis. 

Finally, while LPA as a dimension-reduction approach offers more flexibility in cluster assignment relative to ‘hard clustering’ techniques (i.e., K-means clustering), it can leave uncertainty in latent class assignment [50,51]. When we accounted for this uncertainty in latent class assignment in the sensitivity analysis, it did not materially affect the precision of the findings [95]. We accounted for uncertainty in latent profile assignment using a three-step approach for estimation of covariate effects, in which we (i) constructed our latent profile model, (ii) calculated latent profile assignments, and (iii) related covariates of interest to latent profiles. This approach has been known to underestimate the covariate–profile association compared with single-step approaches. Future work may employ diattenuation approaches to correct the final estimates [51,95].

## 5. Conclusions

In this cohort, maternal folate status or folic acid intake did not modify the association between gestational urinary DMA concentrations and child neurodevelopmental or anthropometric health at age 3 years. We observed suggestive evidence that urinary DMA levels were associated with decreased likelihood of optimal versus typical neurodevelopment. Future work could consider a phenome approach for classifying multiple child-health outcomes, and examine the potential for maternal folate status to modify the relations of phenotypic profiles with gestational or early childhood exposure to arsenic, in populations with higher arsenic exposure or lower maternal folate.

## Figures and Tables

**Figure 1 ijerph-19-11332-f001:**
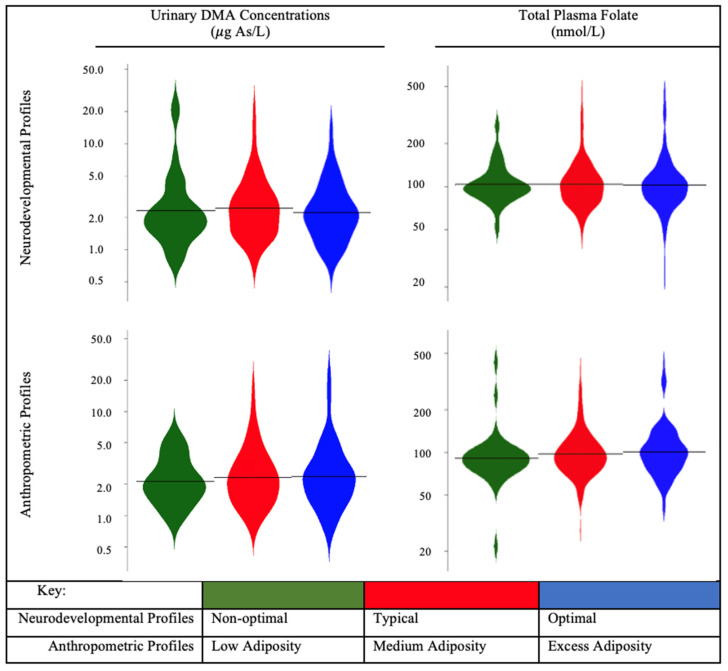
Violin plots of first trimester maternal urinary concentrations of DMA and total plasma folate during pregnancy by neurodevelopmental and anthropometric phenotypic profiles among MIREC study participants; MIREC: Maternal–Infant Research on Environmental Chemicals; DMA: dimethylarsinic acid; total folate: sum of five folate vitamers, 5-methylTHF, folic acid, tetrahydrofolate (THF), 5-formylTHF, and 5,10-methenylTHF. Horizontal bar represents median value of first trimester urinary DMA concentrations or plasma total folate. For descriptive information of phenotypic profiles, see Appendix A.

**Figure 2 ijerph-19-11332-f002:**
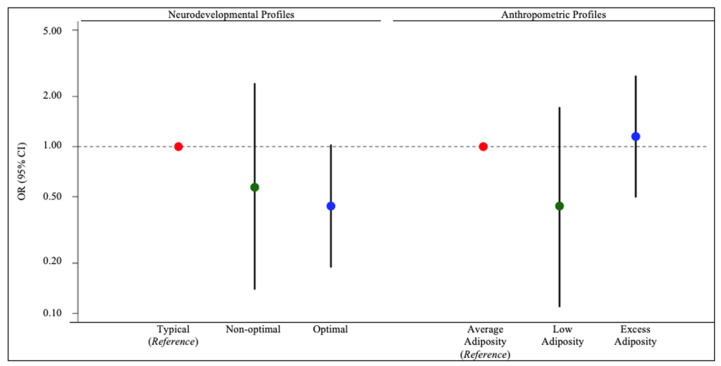
Adjusted associations of neurodevelopmental and anthropometric phenotypic profiles with maternal first trimester log_10_-transformed urinary DMA concentrations (μg As/L) among MIREC study participants. MIREC: Maternal–Infant Research on Environmental Chemicals; DMA: dimethylarsinic acid. For descriptive information of phenotypic profiles, see Appendix A. Models adjusted for maternal age at delivery (continuous), maternal birth country (Canada or foreign), prenatal multivitamin use (yes or no), maternal depressive symptoms (continuous CES-D scores), maternal pre-pregnancy BMI (normal/underweight or overweight or obese), income (continuous), and parity (nulliparous, one previous child, two or more previous children). Error bars represent 95% confidence intervals. Note, the y-axis is on the log scale.

**Figure 3 ijerph-19-11332-f003:**
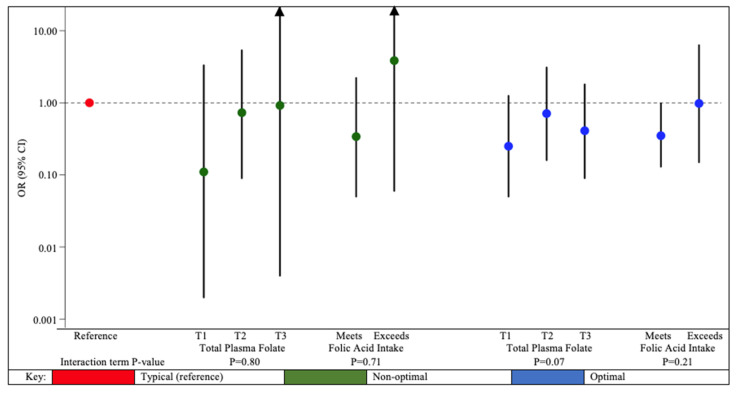
Adjusted odds ratios of neurodevelopmental phenotypic profile membership with 10-fold increase in maternal first trimester urinary DMA concentrations, stratified by terciles of total plasma folate and categories of folic acid intake from supplements among MIREC study participants. MIREC: Maternal–Infant Research on Environmental Chemicals; DMA: dimethylarsinic acid; BMI: Body mass index. Total folate: sum of five folate vitamers, 5-methylTHF, folic acid, tetrahydrofolate (THF), 5-formylTHF, and 5,10-methenylTHF. For descriptive information of phenotypic profiles, see Appendix A. Models adjusted for maternal age at delivery (continuous), maternal birth country (Canada or foreign), prenatal multivitamin use (yes or no), maternal depressive symptoms (continuous CES-D scores), maternal pre-pregnancy BMI (normal/underweight or overweight or obese), income (continuous), and parity (nulliparous, one previous child, two or more previous children). Total plasma folate concentrations were stratified into terciles, with more information available in Appendix A. Due to small sample size we excluded participants who did not meet daily recommendations for folic acid intake from supplements, resulting in a subsample of *n* = 345 participants, see Appendix A for additional details. Error bars represent 95% confidence intervals. Note that the y-axis is on the log scale.

**Figure 4 ijerph-19-11332-f004:**
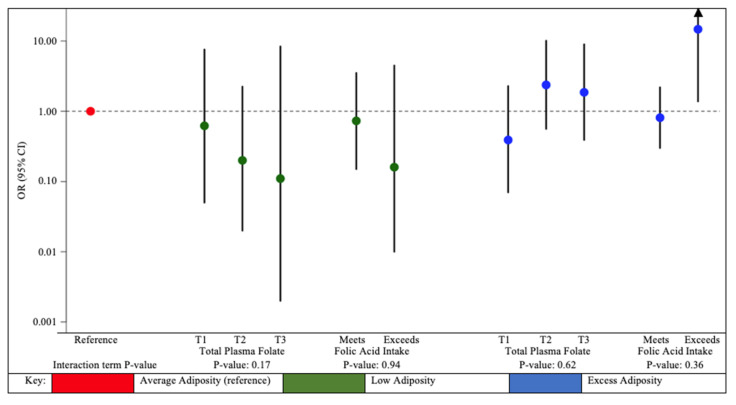
Adjusted odds ratios of anthropometric phenotypic profile membership with 10-fold increase in maternal first trimester urinary DMA concentrations, stratified by terciles of total plasma folate and categories of folic acid intake from supplements among MIREC study participants. MIREC: Maternal–Infant Research on Environmental Chemicals; DMA: dimethylarsinic acid; BMI: Body mass index. Total folate: sum of five folate vitamers, 5-methylTHF, folic acid, tetrahydrofolate (THF), 5-formylTHF, and 5,10-methenylTHF. Daily folic acid intake from supplements was based on maternal self-report using standardized questionnaires. For descriptive information of phenotypic profiles, see Appendix A. Models were adjusted for maternal age at delivery (continuous), maternal birth country (Canada or foreign), prenatal multivitamin use (yes or no), maternal depressive symptoms (continuous CES-D scores), maternal pre-pregnancy BMI (normal/underweight, overweight, obese), income (continuous), and parity (nulliparous, one previous child, two or more previous children). First trimester total plasma folate concentrations were stratified into terciles, with more information available in Appendix A. Due to small sample size, we excluded participants who did not meet daily recommendations for folic acid intake from supplements, resulting in a subsample of *n* = 345 participants, see Appendix A for additional details. Error bars represent 95% confidence intervals. Note that the y-axis is on the log scale.

**Table 1 ijerph-19-11332-t001:** Maternal first trimester urinary DMA concentrations and total plasma folate concentrations by sociodemographic and perinatal characteristics among MIREC study participants.

	Full Sample	Urinary DMA (μg As/L)	Total Plasma Folate ^a^ (nmol/L)
Variable	N (%)	Median (1st Q, 3rd Q)	Median (1st Q, 3rd Q)
	364 (100)	2.23 (1.53, 3.38)	99 (81, 120)
Maternal age (years)			
<30	81 (22)	2.00 (1.41, 3.01)	91 (78, 113)
30–<35	155 (43)	2.25 (1.50, 3.25)	100 (83, 118)
≥35	128 (35)	2.47 (1.59, 3.75)	99 (81, 122)
Education ^b^			
High school or less	12 (3)	1.83 (1.83, 3.42)	97 (84, 126)
Some college	11 (3)	2.31 (2.31, 3.46)	91 (80, 198)
College/trade school	85 (23)	2.06 (2.06, 2.85)	100 (79, 124)
University degree	254 (70)	2.37 (2.37, 3.70)	100 (82, 120)
Maternal birth country			
Canadian-born	304 (84)	2.19 (1.52, 3.19)	99 (81, 118)
Foreign-born	60 (16)	2.48 (1.57, 4.18)	97 (80, 122)
Household income			
<$70,000	86 (24)	2.06 (1.43, 3.17)	95 (75, 111)
$70,000–<$100,000	117 (32)	2.20 (1.53, 3.18)	100 (80, 124)
≥$100,000	161 (44)	2.45 (1.59, 4.00)	100 (83, 122)
Plasma cotinine ^c^			
Unexposed	254 (70)	2.20 (1.54, 3.40)	99 (81, 119)
Second-hand smoking	100 (27)	2.43 (1.51, 3.61)	101 (80, 120)
Active smoking	10 (3)	1.83 (1.43, 2.14)	85 (78, 129)
Pre-pregnancy BMI			
Normal/underweight	226 (62)	2.30 (1.56, 3.37)	95 (80, 114)
Overweight	79 (22)	2.17 (1.54, 3.87)	104 (82, 129)
Obese	59 (16)	2.12 (1.40, 2.96)	99 (88, 119)
Parity			
0	174 (48)	2.35 (1.59, 3.68)	103 (82, 125)
1	150 (41)	2.12 (1.47, 3.18)	98 (79, 114)
≥2	40 (11)	2.28 (1.49, 3.23)	92 (78, 117)
Prenatal vitamin use			
Yes	326 (90)	2.25 (1.55, 3.41)	100 (82, 122)
No	38 (10)	1.86 (1.40, 3.27)	83 (70, 97)
Folic acid supplementation ^d^			
0–400 (μg/day)	19 (5)	3.77 (2.77, 5.56)	80 (71, 106)
≥400–1000 (μg/day)	254 (70)	2.14 (1.51, 3.18)	98 (80, 118)
>1000 (μg/day)	91 (25)	2.43 (1.52, 3.35)	104 (87, 122)
Child sex			
Girls	178 (49)	2.26 (1.55, 3.19)	100 (80, 123)
Boys	186 (51)	2.13 (1.49, 3.65)	98 (81, 117)

MIREC: Maternal–Infant Research on Environmental Chemicals study; DMA: dimethylarsinic acid; Q: quartile (first and third quartile refer to the 25th and 75th percentile); BMI: body mass index; ^a^ Total folate: sum of five folate vitamers, 5-methylTHF, folic acid, tetrahydrofolate (THF), 5-formylTHF, and 5,10-methenylTHF. ^b^ Note, values do not sum to 364, as we did not exclude participants with missing data on maternal education, as this was not a covariate adjusted for in our models. ^c^ Maternal smoking during pregnancy estimated based on maternal plasma cotinine concentrations during the first trimester of pregnancy. Values ≤ 0.15 ng/mL were considered unexposed, >0.15–3.0 ng/mL as second-hand smoking, and >3.0 ng/mL as active smoking [47]. ^d^ Daily folic acid intake from supplements was based on maternal self-report using standardized questionnaires. Supplementary intake was assessed once after the first trimester baseline visit. 0 includes participants who indicated that they did not take folic acid supplements.

## Data Availability

The MIREC study data is available for use by other investigators pending approval by the MIREC Biobank Committee (http://www.mirec-canada.ca). Interested parties can contact Nicole Lupien (nicole.lupien@recherche-ste-justine.qc.ca), the Biobank Manager at to request application materials.

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
