# Peer review of "Maternal Folate Status and the Relation between Gestational Arsenic Exposure and Child Health Outcomes"

_ijerph, 2022, doi:10.3390/ijerph191811332_

Round 1
Reviewer 1 Report
The provided research is very interesting and presented in a well-structured manner. It is relevant for the field and it is original.
The article is written appropriately. Furthermore, the methods, tools, and software are meticulously highlighted in the paper and the results are following the details given in the methods section.
It may be useful to include pregnancy pathology in the model.
The figure captions are thoroughly descriptive, something that negatively affects the layout.
Please review table 1 to be within the limits. In addition, the variable "education" does not sum to 364.
Moreover, some typos need to be corrected (e.g. the title of table 1, on line 846, etc.)
The reference list is conformed to its reference style and is indeed the most recent one. Additionally, cited references in the text should appear before the full stop at the end of each sentence.
All in all, the work fits the journal scope and therefore, the article is worthy of publishing.
Author Response
Comments and Suggestions for Authors
The provided research is very interesting and presented in a well-structured manner. It is relevant for the field and it is original. The article is written appropriately. Furthermore, the methods, tools, and software are meticulously highlighted in the paper and the results are following the details given in the methods section. All in all, the work fits the journal scope and therefore, the article is worthy of publishing.
- It may be useful to include pregnancy pathology in the model.
We agree that including pre-eclampsia, gestational diabetes, or other pregnancy induced hypertensive disorders may be important covariates to consider in adjusted models, based on their association with maternal and child health outcomes. However, we did not adjust for these covariates in our models given that we would not have been well-powered to conduct these analyses. Specifically, only 7.3% of pregnant women developed gestational hypertension without pre-eclampsia, 2.8% developed pre-eclampsia and up to 8% had some form of glucose intolerance.12,3 We note that excluding them would have reduced the generalizability of our findings. However, we included this as a limitation of our work and provided suggestions for future directions in the limitation section of the discussion (lines 542-548).
- Borghese MM, Walker M, Helewa ME, Fraser WD, Arbuckle TE. Association of perfluoroalkyl substances with gestational hypertension and preeclampsia in the MIREC study. Environ Int. 2020 Aug;141:105789. doi: 10.1016/j.envint.2020.105789. Epub 2020 May 11. PMID: 32408216.
- Shapiro GD, Dodds L, Arbuckle TE, Ashley-Martin J, Fraser W, Fisher M, Taback S, Keely E, Bouchard MF, Monnier P, Dallaire R, Morisset A, Ettinger AS. Exposure to phthalates, bisphenol A and metals in pregnancy and the association with impaired glucose tolerance and gestational diabetes mellitus: The MIREC study. Environ Int. 2015 Oct;83:63-71. doi: 10.1016/j.envint.2015.05.016. Epub 2015 Jun 20. PMID: 26101084.
- Ashley-Martin J, Dodds L, Arbuckle TE, Bouchard MF, Shapiro GD, Fisher M, Monnier P, Morisset AS, Ettinger AS. Association between maternal urinary speciated arsenic concentrations and gestational diabetes in a cohort of Canadian women. Environ Int. 2018 Dec;121(Pt 1):714-720. doi: 10.1016/j.envint.2018.10.008. Epub 2018 Oct 12. PMID: 30321846.
- The figure captions are thoroughly descriptive, something that negatively affects the layout.
The figure captions include sample size and descriptions of the Neurodevelopmental and Anthropometric phenotypic profiles. This information adds a substantial amount of text to the captions. To reduce the length of the figure captions, we reference Supplemental Table S1. To reduce the length of the captions for Figure 3 and Figure 4, we reference Supplemental Table S8 and S9 for further details on the terciles of total plasma folate concentrations and the changes in sample size within strata of folic acid intake.
- Please review table 1 to be within the limits. In addition, the variable "education" does not sum to 364.
Please note that we will upload separate documents for Tables and Figures, as to avoid the possibility of distortion by including them in the manuscript document. Table 1 is formatted such that it is contained to one page.
Regarding maternal education, we did not exclude participants with missing data on maternal education since this was not a covariate we adjusted for in our models (see DAG, supplemental Figure S2). We included a footnote to Table 1 to avoid confusion.
- Moreover, some typos need to be corrected (e.g. the title of table 1, on line 846, etc.)
We have addressed the typo for the title of Table 1 (removed the repeated phrasing). The manuscript has also been re-reviewed for grammatical issues.
- The reference list is conformed to its reference style and is indeed the most recent one. Additionally, cited references in the text should appear before the full stop at the end of each sentence.
We have updated the references to ensure the style is consistent with journal requirements and that the reference list is the most recent version.

Reviewer 2 Report
dear authors
thank you for submitting your work to ijerph journal
we have read the manuscript and we have a few suggestions to increase the impact of the work ,we attached a revised file with all the notes marked
we hope it will increase the impact of your article
generally speaking ,English proofreading is recommended
the abstract methods and conclusion needs revision
the introduction background is deficient needs to be revised with emphysisi on important of the subject and the way that folate and arsinic compounds inntract
the ration of the study and why you choose the first trimester is missing too
references from 11-18 were also missing and not cited in the text
methods were missing the study type and clear exclusion criteria .
sample size was not calculated
over references was noted in the introduction and in the methods section
result was over-expresed and repeated too many times ,pleases revise
discussion
lacked the comparison with earlier study and the evidence that explain or back up the reached results
limitation of folate measurement and confounder that affect its estimation as well as arsinic compound estimation was missing too
reference
please up date old references

Author Response
Comments and Suggestions for Authors
dear authors, thank you for submitting your work to ijerph journal. we have read the manuscript and we have a few suggestions to increase the impact of the work ,we attached a revised file with all the notes marked. we hope it will increase the impact of your article
- Generally speaking ,English proofreading is recommended
Multiple native English speakers have thoroughly reviewed the manuscript for grammar, punctuation, etc.
- the abstract methods and conclusion needs revision
Please see below for specific responses to comments left in the manuscript document by section.
- the introduction background is deficient needs to be revised with emphysisi on important of the subject and the way that folate and arsinic compounds inntract
We agree that describing the way in which folate and arsenic interact is an important piece of this work. While we do describe this briefly in the introduction, we go into greater detail in the discussion section. It was our goal to keep the introduction brief, and to avoid repetition in the discussion section.
- the ration of the study and why you choose the first trimester is missing too
While plasma folate status was collected during the 1st and 3rd trimester, we only considered 1st trimester values here. The rationale is described in the methods section (lines 120-21).
- references from 11-18 were also missing and not cited in the text
This was an error in refreshing the citation manager, and this has been resolved in the manuscript.
- methods were missing the study type and clear exclusion criteria .
We used a prospective cohort study, as sated in the abstract (lines 18-19), introduction (lines 65-68), and the methods sections (lines 73-76).
Exclusion criteria specific to the MIREC Study cohort are briefly described in the methods section (lines 77-81), with references provided to the original cohort profile. To obtain our analytic sample, we describe exclusion criteria in the methods (lines 81-91), and provide additional information in a flow diagram (Supplemental Figure S1).
- sample size was not calculated
We present a flow diagram (Supplemental Figure S1) outlining how we obtained our analytic sample size.
- over references was noted in the introduction and in the methods section
Regarding the introduction, we have reduced the number of references while also providing adequate citations for our written statements and attribution to original works.
Regarding the methods section, our analytic plan and methods were guided by findings from prior work. Thus, we do not feel it would be appropriate to exclude these references. We include references to specific methods to provide justifications for the analyses we conducted.
- result was over-expresed and repeated too many times ,pleases revise
We apologize if it appears that the results are repetitive. We note that Figure 2, Figure 3, and Figure 4 are distinct results and each figure corresponds to different analyses.
Results from Figure 2 report the odds of Neurodevelopmental and Anthropometric Phenotypic Profile assignment per 10-fold increase in maternal urinary DMA concentrations.
Results from Figure 3 report the odds of Neurodevelopmental Phenotypic Profile assignment per 10-fold increase in maternal urinary DMA concentrations, across terciles of plasma total folate, and by categories of folic acid supplementation.
Results from Figure 4 report the odds of Anthropometric Phenotypic Profile assignment per 10-fold increase in maternal urinary DMA concentrations, across terciles of plasma total folate, and by categories of folic acid supplementation.
- discussion lacked the comparison with earlier study and the evidence that explain or back up the reached results
In the discussion section, we compare our findings to prior work on lines 560-565. We describe the literature on arsenic as a pervasive environmental exposure impacting child health (the basis for our study) from lines 503-536, and literature outlining the use of folate to reduce arsenic related public health burdens from lines 554-559.
- limitation of folate measurement and confounder that affect its estimation as well as arsinic compound estimation was missing too
We include a discussion of folate measurement (lines 647-659) and arsenic exposure measurement error (lines 603-621) in the limitations section.
- please up date old references
We have updated our reference list.
- Comments from manuscript document: Abstract
- your study included 2 aim the relation with neurological development and with obesity .Revise abstract accordingly
- In lines 17-18 we state that we evaluated associations between gestational arsenic exposure and child health. We specify profiles of neurodevelopment and anthropometry in lines 24-25.
- your study type is missing
- See above, we used a prospective cohort study.
- revise you conclusion and bring a clear home tacking
- We revised our conclusion to summarize our main findings.
- Comments from manuscript document: Introduction
- an English editing is recommended
- See comments from above.
- correct typho in fonts the fullstop comes after the [ ].
- See comments from above.
- add missing referees 12-17
- See comments from above.
- [lines: 43-45] this point needs to be address in more depth and the harmful effect
- We added clarification that the mentioned environmental exposures are associated with atypical neurodevelopment.
- you mean children born to mothers with higher foliate? You
- We clarified that this is among children whose mothers had higher folate.
- Comments from manuscript document: Materials and Methods
- overall the method was intensively described ,revision is needed for some of the basic test and examination that are already described else where .
- We have reviewed and revised the text to avoid repetition.
- the study type is missing ,exclusion criteria and how was missing data manged needed to be clarified
- See above, we identified that this was a prospective cohort study.
- MIREC study specific exclusion criteria are described, and a flow diagram is used to determine exclusion criteria leading to the analytic sample.
- was there an gae limit to maternal age? since 35 years and older are risky of neural tubes disease infants and high BMI too
- Participants were not excited from the MIREC sample or our analytic sample based on maternal age. To account for maternal age and pre-pregnancy BMI, we adjusted for these factors in our analyses (see Supplemental Figure S2 for more information on covariates adjusted for in models).
- we see that initaially the number was big then it reduced to 3 uneven groups
- We used latent profile analysis (LPA) to identify phenotypic profiles. These methods are discussed in section 2.4.
- was sample size calculated ?
- See comments from above.
- there is no need for this high reference in the methods
- See comments from above.
- did you excluded cases of morbid obesity ? or did you
- See comments from above. We did not exclude participants with obesity, however we did adjust for maternal pre-pregnancy BMI. We note that morbid obesity is not used as a clinical term, rather obesity is characterized into subcategories (Class 1: BMI 30-<35, Cass 2: BMI 35-<40, Class 3: BMI >=40).
- what are you exclusion criteria regarding confonders
- We excluded participants who did not have complete covariate data on confounders we adjusted for (see methods sections lines 89-91). We identified covariates to adjust for in our analyses using a directed acyclic graph (DAG), with methods described (lines 244-2249). Our DAG is reported in the supplement (Supplemental Figure S2)
- why did you choose 1st trimester and not both?
- See comments from above.
- this is over citation revise please
- See comments from above.
- what is the need for reference here? [ line 211-212]
- See comments from above. Specifically, these references direct the reader to a prior study from the MIREC platform that validated use of cotinine concentrations to tobacco smoke exposure status, and a second study that identifies optimal serum cotinine cut points to identify the differences between unexposed, second-hand smoking, and active smoking.
- over repetition will make reader lose Intreset ,you define your variable here .in the results too and in the table
- See comments from above.
- at which level sensitivity was set in your test?
- We did not rely on significance testing for interpretation of our findings. Additionally, we did not restrict our reporting to significant results. We reporting findings based on the magnitude and direction of the effect measures. This style of reporting is consistent with the recommendations of the American Statistical Association and other epidemiologists that we should not conclude there is 'no association' just because a confidence interval includes the null value of 0.
- Amrhein V, Greenland S, McShane B. Scientists rise up against statistical significance. 2019;567(7748):305-307.
- Wassersteina R, Lazara N. The ASA's Statement on p-Values: Context, Process, and Purpose The American Statistician. 2016;70(2):129-133.
- Comments from manuscript document: Results
- [Table 1] defin abbrviation then use them
- Definitions for all abbreviations in tables and figures are already included in the legend.
- [Figure 1] were there a statistical differance or not? is there a P -value?
- See comment above regarding p-values. We describe the results pertaining to Figure 1 between line 319-322.
- [Figure 2] over repetition should be avoided revise please
- See comments from above.
- [Figure 3] same note
- See comments from above.
- [Figure 4] revise repeated text
- See comments from above.
- Comments from manuscript document: Discussion
- you did not discuss your results in depth ,nor compred with other study in the feild ,you did not show how your finding will affect our current knowledge nor you explain the implication of these result in our recommendation for follow pregnant women many of the questions raised were left with out answer
- See comments from above.
- [lines 437-446] this highlighted section refers to the important of arsine exposure was not discussed in your questionnaire nor in your study parameters it belong to the background of your introduction revise please
- We did not evaluate arsenic exposure via questionnaires, rather we used urinary concentrations of arsenic species evaluated from the 1st trimester of pregnancy. The paragraph in question goes into more depth for the literature evaluating the impact of gestational arsenic exposure on child neurodevelopmental and anthropometric outcomes. We felt this was warranted in the discussion, as we did not cover this in detail in the introduction in order to keep that section brief, and to avoid repetition.
- [lines 447-456] from line 447-456 this is general talk not a discusion relate them to your result or delete please
- We added additional text to this paragraph.
- [lines 458-466] this highlighted text is your study rationale it belong to the introduction revise please and corelate thsee to your results
- Please see above. This information is briefly included in the introduction, and we expand upon it in the discussion to avoid repetition and to keep the introduction section succinct.
- [lines 474-478] you should be able to give a conclusion based on your study ,please provide one and explain the cause of failure of your hypothesis
- We added information to this section, stating we speculate these findings could be attributed to low levels of arsenic and high levels of folate.
- [line 493] an other limitation is the effect of specific forms of mercury could not be investigated. See Lee, B.E., et al., 2010. Interaction between GSTM1/GSTT1 polymorphism and blood mercury on birth weight. Environ. Health Perspect. 118, 437–443. Kim, B., Shah, S., Park, H.S., Hong, Y.C., Ha, M., Kim, Y., Kim, B.N., Kim, Y. and Ha, E.H., 2020. Adverse effects of prenatal mercury exposure on neurodevelopment during the first 3 years of life modified by early growth velocity and prenatal maternal folate level. Environmental Research, 191, p.109909.
- We agree that arsenic is just one of many toxic metals associated with adverse child health outcomes. However, we do not feel it is a limitation of our study that we only considered arsenic (versus mercury, cadmium, lead etc.). In fact, our study design is specific to arsenic, as folate is not involved in the metabolism of mercury.
- Comments from manuscript document: References
- up date old reference
- See comments from above.
- up date old reference
- you did not discuss your results in depth ,nor compred with other study in the feild ,you did not show how your finding will affect our current knowledge nor you explain the implication of these result in our recommendation for follow pregnant women many of the questions raised were left with out answer
- [Table 1] defin abbrviation then use them
- We did not rely on significance testing for interpretation of our findings. Additionally, we did not restrict our reporting to significant results. We reporting findings based on the magnitude and direction of the effect measures. This style of reporting is consistent with the recommendations of the American Statistical Association and other epidemiologists that we should not conclude there is 'no association' just because a confidence interval includes the null value of 0.
- overall the method was intensively described ,revision is needed for some of the basic test and examination that are already described else where .
- an English editing is recommended
- your study included 2 aim the relation with neurological development and with obesity .Revise abstract accordingly

Round 2
Reviewer 2 Report
dear authours
thank you for you submision
though major points was addresed
still 2 important points was not resolved
exclusion criteria
which mean what are ghe basis of exclusion pitentdin the flow chart
you only mentioned 45 in text what about the rest
the cases that affect mental develpomany should br also excluded like thriof diseas did you did ghat? pleas note this is the second time we ask and no adquent responcr is recived
Sample size it means you calculaye how much cases will be enough to represent community under study
done by Epi statistics, Spps or how to calculate sample in medical reserch
please revise the text and add this to the end of your methods
